# “A Sustained, Productive, Constructive Relationship with Someone Who Can Help”—A Qualitative Exploration of the Experiences of Help Seekers and Support Persons Using the Emergency Department during a Suicide Crisis

**DOI:** 10.3390/ijerph181910262

**Published:** 2021-09-29

**Authors:** Demee Rheinberger, Diane Macdonald, Lauren McGillivray, Myfanwy Maple, Michelle Torok, Alexandra Nicolopoulos, Fiona Shand

**Affiliations:** 1Black Dog Institute, University of New South Wales, Randwick, NSW 2031, Australia; d.macdonald@blackdog.org.au (D.M.); l.mcgillivray@blackdog.org.au (L.M.); m.torok@unsw.edu.au (M.T.); a.nicolopoulos@blackdog.org.au (A.N.); fionas@unsw.edu.au (F.S.); 2Faculty of Medicine and Health, University of New England, Armidale, NSW 2351, Australia; mmaple2@une.edu.au

**Keywords:** suicide, emergency department, qualitative methods, help seekers, support persons, carers, lived experience

## Abstract

For Australians experiencing a suicide crisis, the emergency department (ED) is the recommended point of contact for intervention and to ensure personal safety. However, negative ED experiences can deter individuals from returning, thus impacting future suicide risk. In order to improve the ED environment for individuals in suicidal crisis, an in-depth understanding of this experience is needed. In-depth semi-structured interviews with 17 help seekers and 16 support persons were conducted. A grounded theory approach uncovered a core organising concept—all participants wanted a “a sustained, productive, constructive relationship with someone who can help” during the ED visit—which guided analysis. Thematic analysis resulted in two themes and four subthemes exploring the systemic and interpersonal aspects of the ED visit and the roadblocks and pathways to development of the relationship. Interpersonal factors included aspects of staff interaction and presence of a support person. Systemic factors related to aspects controlled by the physical space and internal policies and procedures and included aspects such as the chaotic environment, long waiting times, and access to staff. Overwhelmingly, there were more roadblocks than pathways reported by participants. Improving the ED environment, increasing staff training and encouraging the presence of support persons may help mitigate some of these roadblocks.

## 1. Introduction

For Australians experiencing a suicide crisis, both after a suicide attempt or when experiencing severe suicidal ideation, the emergency department (ED) is frequently the point of contact for intervention with particular regard to personal safety. However, the majority of ED presentations during 2019 and 2020 were for physical concerns [1], so inevitably ED processes and environments have been designed with less focus on support for psychological distress. A negative ED experience for individuals in a suicide crisis presenting to EDs may reduce their willingness to return to the ED in future suicidal crises [2], which is concerning as people with a previous suicide attempt are at increased risk of dying by suicide—particularly in the first 12 months following an attempt [3]. Between 14–22% of individuals who present to hospitals with a previous suicide attempt will make another attempt within one year of presentation, of which 1.5% to 3% are fatal [4,5]. It is vital that appropriate and effective suicide crisis care is available and used in all EDs, or is considered in the development of ED alternatives for suicidal crisis.

This study is an exploratory investigation of the experiences of suicidal presentations to the ED from two perspectives: (i) help seekers—people who present to an ED with suicidal thoughts or a suicide attempt, and (ii) support persons—loved ones or unpaid carers who attend an ED with a help seeker. These terms were determined through consultation with Black Dog Institute’s Crisis and Aftercare Lived Experience Advisory Group (CALEAG). The term ‘suicidal crisis’ is employed as an all-encompassing term for suicidal behaviour (attempts) and suicidal thoughts (ideation). 

The ED can be a challenging environment for help seekers as a place to seek care. Help seekers have indicated that the chaotic and fast-paced environment of the ED, combined with long waiting times, is detrimental to their vulnerable state of mind [6,7]. Negative staff attitudes, such as low empathy, and a focus on physical injury have resulted in help seekers reporting negative ED experiences [2,6,7,8]. Help seekers have also reported that their needs are not being met by the ED staff, with no treatment provided for their presentation and/or their presenting problem being dismissed [6,7].

A large portion of help seekers are prompted to attend the ED by someone in their support network, be that family, friends, or a member of their health care team [6,8,9], and this person often accompanies them during this experience. The presence of support persons at the ED can help to mitigate the emotional distress experienced whilst in hospital and improve communication between help seekers and healthcare providers [10]. However, little is known about the experience of support persons when accompanying a suicidal individual to the ED. Support persons find it challenging to keep a loved one safe during a suicidal crisis [11,12] and often feel as if they are not adequately equipped to take on the responsibility of caring for an individual during this time [13,14]. Despite being an integral part of the help seeker’s care, support persons have reported feeling ED staff do not view them as essential team members in the treatment and care of suicidal individuals [11,14].

The experience of suicidal crisis is alarming and stressful for both the individual and the support persons around them. A visit to the ED is only one step in an individuals’ pathway to treatment, yet it often appears at a critical junction—when the person is in acute crisis. The first 30 days following a visit to the ED for a suicide crisis are when help seekers are at greatest risk or reattempt or death [15], as such, the ED is uniquely placed to assist help seekers and their support persons to adequately prepare to continue with treatment and care once the acute crisis has been resolved. . Yet often EDs do not provide this support or assistance to individuals visiting EDs for suicidal crisis [16,17].

To ensure that EDs are an appropriate place for individuals in suicidal crisis to both manage a crisis and prepare them for ongoing treatment and care, it is important to understand help seekers’ and support persons’ lived experience of using an ED during a suicidal crisis. Qualitative research methods provide a unique opportunity to gain a rich, contextual understanding of lived experience, where researchers can gain insights of help seekers’ and support persons’ experiences in an exploratory, yet meaningful way. It is essential as we move toward improving quality of care in EDs, and creating viable alternatives to the ED, that the experiences of help seekers and support persons are considered to ensure that service delivery aligns with their needs.

To date, the authors are not aware of any previous research that qualitatively explores the perspectives of both help seekers and support persons who have experience of using an Australian ED during a suicide crisis. The aim of this study is to understand the experiences of help seekers and support persons during a suicide-related ED presentation to identify from their perspectives, what works, what does not work, and areas for improvement in the delivery of crisis care for suicide in Australia. The knowledge drawn from this study can be used to inform evidence-based reforms to crisis care, including improvements to existing ED services and the development of alternatives to the ED.

## 2. Method

This study was approved by the Hunter New England Human Research Ethics Committee (HREC/17/HNE/144). Participants were provided with participant information sheets prior to their interview. Verbal or written consent was obtained from all participants prior to the commencement of interviews.

This research uses qualitative data from a longitudinal cohort study [18], which forms part of a larger suicide prevention trial, known as LifeSpan [19]. The CALEAG was consulted during the development of the interview guides to ensure the language, topics of investigation, and questions were respectful and representative of the lived experience of suicide. The panel consisted of three people with lived experience of having attended the ED for a suicidal crisis in a help seeker and/or support person capacity. 

### 2.1. Participant Recruitment

Help seekers (*n* = 17; female = 16) and support persons (*n* = 16; female = 11) attending the ED for suicidal crises were recruited from six Australian Local Health Districts (LHDs) in New South Wales (NSW) and the Australian Capital Territory (ACT) between May 2019 and January 2020. Help seekers were recruited through their participation in the cohort study survey: after participants completed a survey, they were invited to take part in semi-structured interviews. Support persons were recruited either through referral from help seekers or via Black Dog Institute’s social media platforms (Facebook and LinkedIn) between October 2019 and January 2020.

Participants were 16 years of age or older, who self-identified as a help seeker or support person who had presented with a suicidal crisis (attempt or thoughts) to an ED in one of the six study regions in the 18 months prior to the interview. These two samples did not necessarily need to present at the ED together, at the same time and/or know each other. Any support person who had a loved one (help seeker) subsequently die by suicide was precluded from the study. Support persons were either a parent (*n* = 10), current or ex-partner/spouse (*n* = 4), friend (*n* = 1) or other relative (*n* = 1) of the presenting help seeker. There were three occurrences whereby a help seeker and support person were directly linked, as such, sharing their unique experience of the same ED visit. 

### 2.2. Data Collection

Semi-structured interviews (*n* = 33) were conducted face-to-face, over the phone, or via video conferencing depending on the participant’s preference. Interview questions were designed to explore the participants’ experiences with the ED and support after discharge (known as aftercare) with emphasis on the following key topics: access to care, assessment, handover processes, stigma, staff attitudes and staff knowledge, continuity of care, discharge processes and care beyond the ED. The interviews were approximately 60–90 min in duration, digitally recorded, and were conducted by six members of the research team. Digital recordings were de-identified and sent to a third-party secure service for verbatim transcription. 

### 2.3. Analysis

Interviews were reflexively analysed using thematic analysis [20] to find patterns and commonalities of participant’s ED experiences through a rigorous process of data interrogation. This was used in combination with a grounded theory approach [21] which was utilized to guide the direction and coding structure of the thematic analysis, and frame meaning around help seeker and support person’s stories through inductive, iterative, and comparative methods. Through this process a core organising concept was uncovered—that both help seekers and support persons wanted “a sustained, productive, constructive relationship with someone who can help”. Data were then examined using Braun and Clarke’s six phases of thematic analysis [20] to determine what influenced the development, or not, of this relationship in the ED. Interviews were read in full, twice, and independently coded line-by-line by three members of the research team. This process resulted in the development of 32 initial themes. Through repeated engagement with the data and robust discussion between authors, themes were honed into a hierarchy of two major themes (interpersonal and systemic) with two sub-themes each (pathways and roadblocks). 

In addition, the CALEAG were engaged throughout the analytical period for advice and validation of our process and findings. Consultation between the research team and the CALEAG throughout this process ensured rigour was maintained and allowed for reflexivity in subjectivities throughout the process as assumptions, findings, and contradictory data were checked.

## 3. Findings

All participants wanted access to “a sustained, productive, constructive relationship with someone who can help” (support person 11, male) and thematic analysis resulted in two themes and four sub-themes which detail how this was experienced (Table 1). Themes explored either systemic or interpersonal elements of the ED experience, which were further broken into sub-themes outlining the roadblocks or pathways to the development of this relationship.

### 3.1. Interpersonal

Interpersonal aspects of the ED visit for individuals in a suicide crisis are those which are created as an interaction that occurs between those presenting to the ED and those who work in the ED. These aspects were determined to be either pathways (elements which facilitated) or roadblocks (elements which hindered) access to a “sustained, productive, constructive relationship with someone who can help”. 

#### 3.1.1. Roadblocks

##### Negative Interactions with ED Staff

Negative staff interactions profoundly impacted help seekers’ and support persons’ ability to attain a sustained, productive, constructive relationship with someone who could help. A lack of compassion and empathy was repeatedly mentioned by both help seekers and support persons: 

“You feel very dismissed by them [ED staff] because you look like there’s nothing physically wrong with you.”—Support person 3, female.

“I felt like I was totally ignored in the emergency department”—Help seeker 11, female.

Some ED staff expressed problematic beliefs and stigmatising views towards individuals presenting to the ED for care, which participants felt detrimentally influenced the quality of care they received:

“Basically [the doctor’s] view was, if I was going to kill myself, I would have left the emergency department to do it.”—Help seeker 10, female.

“I buzzed the nurse, and basically, she said to me that I had done this to myself, it’s all your fault, and blaming and shaming me for what happened,”—Help seeker 17, female.

Some participants detailed experiences that could be seen as being staff misconduct, and were certainly examples of mistreatment:

“I remember she [the nurse] put in a local [anaesthetic]. And then I was quite distressed. And then she started to stitch, and there was a part where it felt she obviously didn’t local enough. And I was like, “I can feel that.” And I got quite upset, and she kind of said … ‘Well, you don’t mind the pain though, do you?’”—Help seeker 5, female.

##### Reliance on Support Persons Initiative to Ensure Care

Support persons were dismayed that using their own initiative in the ED seemed to be essential to ensuring that their loved one was given the care and treatment they needed. 

“… if you are worried, and you don’t think that your loved one or your family member … is getting adequate help, then keep trying and try every avenue and even if you have one door shut, knock on it, bash it down, whatever …”—Support person 1, female.

Help seekers and support persons spoke of instances when this stigma and negative attitude became detrimental to the psychological wellbeing of the help seeker. For support persons this was particularly difficult, they had often brought their loved ones to the ED as the last option when other interventions were failing to keep their loved one safe. A pervasive sentiment, particularly among parents of children with suicidal thoughts or behaviours, was that ED staff felt their children are “*better off at home*” (support person 16, female) with the support person; a sentiment that support persons did not fully agree with.

##### Help Seeker Obstruction of Care from ED Staff

To further complicate this, support persons reported that their loved ones would often lie to reduce the seriousness of the presentation encouraging ED staff to allow them to leave the ED without being seen by a member of the mental health staff. In many instances, the knowledge of what to say to manoeuvre the ED staff into sending the help seeker home prematurely was developed over multiple ED visits. It is interesting to note, that most help seekers who used these tactics were adolescents rather than adult help seekers.

“She’s very smart and it is a manipulation of the rules. It’s like that’s not there, it’s to encourage people and motivate people to go but she then turns it and goes, “I want to get out of here. So that’s what I’m going to do.”—Support person 14, female.

##### Discharge When Out of Crisis

Support persons can find the decision to discharge their loved one distressing. Many reported feeling as though the decision to send their loved one home was based on the degree of crisis the help seeker was experiencing in that moment. However, other support persons felt they were unable to ensure their loved ones’ safety, which was not considered by the staff making the decision to discharge the patient: 

“… the doctor said, ‘Are you happy for him to go home?’ And I said, ‘Well, no, I’m not.’ Then the next thing you know … they just said, ‘He can go home.’”—Support person 2, female.

##### No Support for Support Persons

Participants noted that there was little support available in the ED for support persons. Visiting the ED was described as an overwhelming and scary experience for the support person as well as for the help seeker; both help seekers and support persons in this study recommended that more support be made available for support persons to help manage the distress, understand the process and access respite: 

“Nobody comes and talks to the family and saying, ‘OK, this is what you’re going to do together’ but the expectation is that the person will be looked after at home but there’s no support for the family. There’s only support for the [help seeker].”—Support person 14, female.

#### 3.1.2. Pathways

##### Positive Interactions with ED Staff

Both help seekers and support persons also remarked upon positive interactions in the ED. These exchanges included staff demonstrating respect, empathy and compassion, as well as taking time to be kind to the help seeker or support person:

“She (nurse) was very sympathetic, and empathetic. She was very concerned. It wasn’t like you were just some, somebody that walked in off the street that, you know …?”—Help seeker 1, female.

“a few staff or nurses in particular that made the time to come and just sit and chat for a bit, not necessarily trying to figure out why I’m there, but just talk.”—Help seeker 14, female.

Some help seekers and support persons said that a helpful staff demeanour facilitated meaningful relationship building with ED staff. These actions included speaking softly, being gentle with their physical interactions, and asking thoughtful, conscientious questions during assessments. 

“The fact that they didn’t have a mental health person there, but they still got me in front of someone, and that person was really personable in a scary time.”—Help seeker 20, female.

“When they asked us to wait back in the waiting room, within about an hour … they come back out to check his blood pressure, his temperature. … And later on he said to me … ‘I really liked that, mum, because it made me feel like they hadn’t forgotten me’.”—Support person 8, female.

##### Support Person Involvement in Assessments

Support persons found being involved in the assessments to be a positive aspect of their ED experience noting that staff who were patient and took time to understand their story were well received. They also liked when they were able to speak privately to the assessing doctor or nurse separately from their loved one: 

“… the mental health person … he had a chat with me as well for about half an hour, to get a read on the situation.”—Support person 11, male. 

##### Presence of a Support Person

Most help seekers discussed the importance of having a support person at the ED with them during their ED visit as an advocate for their care when they reported not being able to do so for themselves: 

“After about eight and a half hours she started to push for, why aren’t things happening, you know, why? Whereas I probably would have just sat there.”—Help seeker 1, female. 

“My advice would be make sure … you have someone to advocate for you, because often when you’re in that kind of state, you’re unable to really talk for yourself and really push for your own care.”—Help seeker 17, female.

Help seekers who did not have a loved one with them regretted the decision to go alone and many felt as though a support person would have made the ED experience less distressing. 

“I think part of what made me so upset and so distressed was that I was so alone in it all. And I didn’t have even a friend or family member come with me and kind of stick with me through that, which was a really silly decision. I should have definitely reached out to someone. I would’ve felt so much more comfortable.”—Help seeker 5, female.

### 3.2. Systemic 

Systemic aspects of the ED visit for individuals in a suicide crisis are those which are determined by the physical space, and the policies and procedures that dictate how treatment is delivered in the ED. As with the Interpersonal aspects discussed above, these aspects were determined to be either pathways or roadblocks to accessing a “sustained, productive, constructive relationship with someone who can help”. 

#### 3.2.1. Roadblocks

##### Chaotic ED Environment

The physical environment of the ED is usually noisy, bright, busy and crowded which can be difficult for individuals who are experiencing suicidal thoughts. The presence of other highly distressed or injured patients can exacerbate suicidal ideation or psychological distress and provoke anxiety. In some cases, the ED environment was so challenging that again help seekers left before receiving treatment: 

“So, sitting out in the main ED area is very traumatic for her because there’s a lot of noise, there’s light, there’s lots of people.… so that can be traumatic, and she’s got to that point and says ‘I just want to leave’, and sometimes we’ve had to leave before we actually get any treatment.”—Support person 4, male.

A lack of privacy is also difficult for help seekers in the ED. The paper curtain used to separate beds in the ED often discouraged help seekers disclosing their issues openly with ED staff: 

“There have definitely been times where I’ve tried not to say as much as I want to say just because, I … worried about other people listening in to what I’m saying. … I’ll try to speak more quieter and … try to hold off a lot of what I want to say.”—Help seeker 6, female.

When help seekers were placed in an isolated room, although affording them privacy, it left some feeling forgotten by ED staff, with many commenting that it felt as though they received less treatment due to the isolated location of their bed. This suggests that while a private place for assessment and while in acute distress might be well regarded, being isolated and left alone by ED staff may have negative impacts. 

A few help seekers and support persons felt that a separate ED for mental health presentations would be beneficial. Participants envisioned these mental health EDs as calmer with less of the sound and busyness typical of existing EDs, where participants would have more privacy, and staff have the capacity to be compassionate and patient. 

“I think for me, what I’d really like to see is an ED for mental health patients. Because to me, a mentally unwell person is walking into the franticness of an ED waiting room is very, very tricky.”—Support person 8, female.

One support person suggested “somewhere that provides that specialist care” (support person 5, female) for individuals during a suicidal crisis would be more appropriate than an ED.

##### Prioritisation of Physical Presentation

The nature of the ED also means that in most cases, physical conditions are prioritised over mental health conditions. One help seeker noticed the stark difference in priority provided to physical as opposed to mental health presented and noted “Hopefully you don’t have to actually try to [sic] suicide to actually get help” (help seeker 2, female).

This can be difficult for an individual whose life is at risk. However, with no physical ailments requiring treatment, help seekers who are out of the crisis stage are often deemed well enough to be discharged:

“But they pulled me aside and the mental health worker said to me quite bluntly … they could only keep me overnight if I was going to go home and kill myself.”—Help seeker 20, female.

##### Long Waiting Times

Long waiting times were detrimental to the mental wellbeing and level of care provided to the help seeker. Both help seeker and support persons considered the waiting times to be “unacceptable” and reported waiting time of up to 24 hours before being seen in the ED, with most reporting waiting times of eight to twelve hours. For some help seekers long waiting times resulted in the individual being able to “ride out” the wave of crisis, however this appeared to support confirmation bias from healthcare staff to not provide more urgent care. For others, the long waiting led to individuals simply leaving the ED without any physical or psychological help or intervention for their suicidal presentation. 

“I was asked twice, was I still feeling okay, but after four hours, I walked out.”—Help seeker 4, male.

##### Understaffing in the ED

An understaffed ED exacerbates the issue of waiting times and perceived quality of care. Both the help seekers and support persons noticed that staff seemed to be either absent or appeared to have been working long hours and considered overworked staff detrimental to the quality of care they received. Despite this, participants showed empathy for the staff, particularly nurses. 

“Other times I suppose it’s quite often being short staffed, and they’re stressed and overworked. I know what’s that like, it’s horrendous, it means you can’t do your job properly.”—Support person 3, female.

“I know they’re limited. I know it’s not their fault that they can’t supply all the care I would need or want.”—Help seeker 13, female.

##### Poor Access to Mental Health Staff in ED

Participants found an absence of specialised mental health staff particularly upsetting. Many had to wait until a weekday to see mental health staff or were sent home without seeing a mental health team member.

“I think it was more just them not having the staff there, them not having like the ability to help me, that kind of made the experience even … the experience upsetting for me.”—help seeker 2, female.

“Imagine if we had specialist mental health ... at every single hospital, 24/7 in ED who can deal with these people as they present, because if the mental health nurse can talk that person down and make sure that they’re OK, then they might be able to go home safely with their carer if they can assess that the carer is capable of taking good care of that person.”—Support person 5, female.

Both help seekers and support persons identified that most ED staff had a poor understanding of mental health, in particular, knowledge about suicide generally or how to speak with individuals experiencing a suicidal crisis: 

“He didn’t know what to do or say in terms of mental health.”—Help seeker 13, female.

Participants commented that staff with limited mental health knowledge were particularly worrisome when there was already poor access to mental health professionals in the ED.

##### Transactional Mental Health Assessments

Mental health assessments were sometimes short “probably ten minutes” (help seeker 14, female), and often were perceived as being transactional, appearing as though the healthcare staff were simply “ticking boxes”. 

“It would be nice if they actually used her history in any way, shape or form, or appear to use it. They may well read it, but it doesn’t appear that they do...”—Support person 4, male.

This resulted in help seekers feeling as though the assessment was insufficient and did not cover important aspects of their presentation and reporting they “felt listened to but … felt like [they were not] being understood” (help seeker 6, female). Some help seekers also discussed the difficulty retelling their story multiple times to different staff members through the ED process with one participant saying it was “hard, reliving that trauma again” (help seeker 9, female).

##### Inadequate Discharge Plans

Treatment plans provided on discharge were viewed by some help seekers as inadequate, with help seekers usually only being provided crisis helpline phone numbers or simply requesting the help seeker contact their general practitioner (GP).

“… there wasn’t even a ‘We’ll arrange to make an appointment to speak to this person’ or something. It was a ‘You go and talk to your GP’, not even knowing whether I had a GP.”—Help seeker 18, female. 

“They give you the, you know, they give you the token card, you know, with access lines on it, or Lifeline’s number on it.”—Help seeker 1, female.

#### 3.2.2. Pathways

##### Detailed Mental Health Assessments

Help seekers spoke positively of detailed mental health assessments, particularly when they were asked many questions, but did not feel rushed to respond. Acknowledgement of the person’s distress and of the circumstances that had led to their suicidal crisis was particularly helpful. This led help seekers to feel that a good understanding of their experience and wellbeing was established. Help seekers valued mental health staff who asked about a range of psychosocial aspects which may have influenced their suicidal thoughts/behaviours: 

“… we discussed what circumstances had led me to feeling suicidal … It was nice having someone just acknowledge that these things are tough and hard and that I was going through something. That was a big relief.”—Help seeker 18, female. 

##### Access to Mental Health Staff

Help seekers reported that speaking with a member of the mental health team, either in-person or through video conference, helped create a positive ED experience. Some help seekers who were offered the video conference option were initially frustrated that they were unable to speak with a mental health professional in person. However, many reported a positive experience using the video conferencing facility, and felt grateful that it had resulted in a significantly shorter waiting time in the ED. 

“I thought it was really great, we felt a lot better going there because we got to speak to someone on Skype, we really felt like we were listened to.”—Help seeker 20, female. 

##### Involvement in Discharge Decisions

Some help seekers reported being involved in the decisions about their care after the ED visit, which many were grateful for irrespective of whether the decision was to be discharged or admitted. 

“… they [nurses] turned around and asked me what I wanted to do. I said I think I just want to go home … so I got a taxi and went home.”—Help seeker 15, female.

“I think I was happy with that decision [to be admitted]. I’m not sure that they would have enforced it if I’d said, no, I want to go home, I don’t know. But I felt sort of a little bit relieved that it wasn’t in my hands like that I didn’t have to be in control of my own safety just for that moment.”—Help seeker 14, female.

## 4. Discussion

This study involved qualitative interviews with help seekers and support persons attending the ED for suicide-related crises to understand what works, what does not work, and areas for improvement from a service user perspective. Our findings resulted in two broad themes which explore aspects of the ED experience: Interpersonal and Systemic. Both themes were broken into smaller elements which explored the roadblocks and pathways toward a “sustained, productive, constructive relationship with someone who can help.” Unfortunately, roadblocks to this relationship far outnumbered the pathways—overwhelmingly there were more roadblocks than pathways for both systemic and interpersonal aspects of the ED experience. Interpersonal factors related mostly to engagement between participants and the ED staff and included aspects of staff interaction, presence of a support person, and degree of support available for support persons. Systemic factors related to aspects controlled by the physical space and internal policies and procedures and included aspects such as the chaotic environment, long waiting times, access to staff, and degree of detail collected during mental health assessments.

Interpersonal roadblocks were often typified by the presence of stigmatising attitudes or an absence of empathy. Help seekers and support persons reported feeling as though negative staff interactions impacted the quality of care they received. Additionally, negative and stigmatising interactions including absence of compassion and empathy may reinforce the negative self-evaluations typically seen in individuals experiencing a suicide crisis, such as low self-worth, a sense of burdensomeness, and hopelessness [22]. Negative interactions with ED staff, particularly those displaying stigma directed towards suicide and those in a suicide crisis have been shown to exacerbate feelings of shame and decreases the likelihood of engaging in future help seeking behaviours [7,9]. ED staff are often the first, and most easily accessed, healthcare providers for individuals in a suicidal crisis, so appropriate interactions with help seekers is vital, especially when there is limited or no access to community services. Greater access to suicide specific education can help staff feel confident in providing basic psychological care (e.g., psychological first aid) alongside treatment of the presentation [23,24,25] and reduce stigma towards individuals in suicide crisis [24], which will likely improve individuals experience of using the ED for suicide related presentations.

Positive interpersonal elements were multifaceted, with the relationship between the help seeker, support person, and ED staff all considered to be a pathway towards a “sustained, productive, and constructive relationship with someone who could help”. A 2013 study found that engaging with the help seeker as an individual deserving respect rather than simply treating the suicidal crisis facilitated help seekers’ positive self-evaluations and reducing feelings of hopelessness [26], and small acts of kindness in the ED can encourage help seekers to feel as though they are being considered more than just their presentation [27]. However, staff in EDs are under incredible pressure to provide high quality care with little resources [6], and as such, many staff focus on providing “treatment” for the condition rather than care for the individual in order to meet the demands of the busy ED setting. 

Additionally, the presence of a support person played an important role in improving the ED experience for help seekers in this study. Support persons were able to act as a companion and advocate when the help seeker was not in an emotional state to do so effectively for themselves. While the authors recognise that there are numerous barriers to support person involvement, such as reluctance to share suicidal experience with others, not having a support network, or concern around confidentiality in the ED [28], we suggest that mental health professionals encourage their clients, where possible, to have a support person accompany them to the ED. Inclusion of support persons has been recognised as one of the successes of the early 1990s mental health reforms in Australia [29]. However, this inclusion is not seen in Australian EDs currently, and support persons are not always considered to be an important part of the team providing care to help seekers by ED staff [11,14]. Additionally, support persons often do not have access to help and assistance they may require to effectively provide support to help seekers, something which is often not assessed for prior to discharging help seekers home to the care of their support person [11,14]. For a support person to effectively engage in care of help seekers post discharge from ED more needs to be done to improve the quality and quantity of assistance provided to support persons, however further investigation is needed to determine what would be beneficial for support persons. 

There were numerous systemic roadblocks to a sustained, productive, constructive relationship with ED staff that could assist help seekers when presenting with a suicide crisis. The chaotic nature of the physical ED environment made the ED presentation difficult for help seekers and, in some cases, lead some help seekers to leave the ED prematurely. The ED environment has repeatedly been found to be challenging for individuals presenting with suicide crises [6,9,16]. Even though EDs may not be currently meeting the needs of people with suicidal distress, they remain an important part of the service landscape and as such reforms are essential. Provided that the ED is designed largely to treat physical conditions, they are not adequately structured or resourced to treat mental health related crises on top of this [30], increased funding for staffing and beds may alleviate the mental health burden on EDs. Additionally, there is a critical need to develop an alternative service to support help seekers in suicidal crisis [31]; one that can provide a calmer environment, staff with more time for mental health assessment and greater access to therapeutic support. However, more work needs to be done to understand how these alternative services can be most effective and integrated with primary and community-based care. Whilst alternatives are under development within Australia, evaluation is necessary to ensure they are appropriate in the Australian context. In the meantime, EDs could partition private waiting areas for mental health presentations, and provide suicide prevention training to ED staff, to increase the likelihood that help seekers stay in and return to the ED during a suicide crisis as this is currently their best option to reduce their risk of suicide attempt or death during this time.

Additionally, poor access to mental health professionals profoundly impacted help seekers and support persons in the ED, as this both increased waiting time and resulted in help seekers feeling as though their suicidal crisis was not taken seriously. Increased presence of mental health professionals in EDs has been shown to improve the experience of the ED for help seekers presenting with mental health concerns by reducing waiting times, increased therapeutic engagement and greater communication with general ED staff [32].

While there were few systemic pathways reported by participants, help seekers found that more detailed mental health assessments in the ED were beneficial, making them feel heard and providing them with hope that steps were going to be taken to help them work through their mental health concerns. However, limited training on mental health and suicide related presentations has been shown to reduce the likelihood that adequate mental health assessments or suicide triaging will be completed by ED staff [33]. The lack of mental health professionals available in the EDs then, is likely to impact the help seeker experience of receiving care. Additionally, general ED staff often report not feeling confident conducting mental health assessments [33] or even simply interacting with help seekers presenting in a suicide crisis [34,35], which is concerning given that they are the primary, and possibly only, health contact for help seekers visiting the ED in suicidal crisis, which is reflected in comments from help seekers about physical health care being prioritised over care for a suicidal crisis. A recent Delphi consensus study found that including comprehensive psychosocial assessments and an opportunity for therapeutic engagement may be best practice in the care of help seekers in suicide crisis [36].

It is important to acknowledge the role systemic factors play in influencing the prevalence of interpersonal roadblocks or pathways. Factors such as inadequate resourcing of ED staff both with and without specialisation on mental health can place pressure on staff to work quickly to provide care to as many patients as possible—reducing the time they have available to provide compassion care or engage in detailed assessments [37]. Additionally, a system wide focus on providing greater access to mental health education, or lack thereof, may impact the prevalence of stigmatising beliefs ED staff hold toward suicide and those experiencing suicide crisis and their confidence to provide care [23,24,25]. ED staff are often working under intense and stressful conditions, providing the highest quality care they can with the limited resources they have available to them. Further investigation is needed to understand the impact of systemic factors on the emergency care system and how these systemic roadblocks may be mitigated to provide higher quality care.

Several limitations should be noted. The data collected relies on memories of participants’ ED visits, which took place up to 12–18 months prior to interviews taking place, however the high emotions being experienced during ED visit potentially enhances memory recall, improving reliability of our findings [38]. In addition, the recruitment method involved participants opting in to take part in an interview, which may have resulted in a biased sample where individuals who had a bad experience may be more likely to want to participate. For instance, some participants had experienced a series of failed attempts to receive help up until their presentation to the ED, which may have negatively coloured their recollections of the visit [2]. As there were only three dyads within the sample, the majority of the sample were unrelated people who attended as either help seeker or support person. A greater proportion of dyads, or where triads with ED staff could be interviewed together would triangulate the data and provide a clearer insight into the ED experience for all involved. Future research should look to examine the triad of experiences when utilising the ED for a suicide crisis. Further to this, there was an overrepresentation of female participants, as well as limited demographics were collected from participants, and several at-risk demographic groups were not specifically targeted (such as LGBTIQ and Aboriginal and Torres Strait Islander participants), and as such is it difficult to apply this experience across all individuals engaging in emergency suicide crisis care. Nevertheless, these findings are consistent with other Australian studies [2,27,39,40] and it is vital that we honour the lived experience of help-seekers and support people in any service reform and design processes.

## 5. Conclusions

Our study found that often the ED experience for individuals in a suicide crisis was not conducive to developing a “sustained, productive, constructive relationship with someone who could help.” Numerous systemic and interpersonal roadblocks negatively impact on the ED experience. More needs to be done to ensure that systemic roadblocks can be reduced to facilitate the generation of more interpersonal and systemic pathways. The presence of a support person could improve a help seekers experience in the ED, which should be encouraged where possible. While increased funding and resources benefit help seekers and support persons, further investment in, and evaluation of, alternatives to the ED is warranted to improve outcomes for help seekers experiencing a suicide crisis.

## Figures and Tables

**Table 1 ijerph-18-10262-t001:** Interpersonal and systemic pathways and roadblocks to care.

Theme	Sub-Theme	Codes
3.1. Interpersonal	3.1.1. Roadblocks	Negative interactions with ED staff Reliance on support persons initiative to ensure care Help seeker obstruction of care from ED staff Discharge when out of crisis No support for support persons
	3.1.2. Pathways	Positive interactions with ED staff Support person involved with assessments Presence of a support person in the ED
3.2. Systemic	3.2.1. Roadblocks	Chaotic ED environment Prioritisation of physical presentation Long waiting times Understaffing in the ED Poor access to mental health staff in ED Transactional mental health assessments Inadequate discharge plans
	3.2.2. Pathways	Detailed mental health assessments Access to mental health staff Involvement in discharge decisions

## Data Availability

Data available upon request from the corresponding author. The data are not publicly available due to privacy restrictions.

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
