# Peer review of "“A Sustained, Productive, Constructive Relationship with Someone Who Can Help”—A Qualitative Exploration of the Experiences of Help Seekers and Support Persons Using the Emergency Department during a Suicide Crisis"

_ijerph, 2021, doi:10.3390/ijerph181910262_

Round 1
Reviewer 1 Report
The study is addressing an important and understudied safety net for suicide crises. The authors are correct in saying that qualitative data is the best approach to understanding the experiences of those seeking help and those supporting them in doing so. The qualitative analysis appears to be rigorous and is described in detail. The manuscript is well-written and interesting to read. This is a well-conducted study. I think the data are reliable due to the emotionally-charged situation the help seekers and support people find themselves in during an ER visit. They are likely to accurately recall what was said to them and how they felt.
Questions that arose during the reading and review of the manuscript are:
- Are there any data on how many people presenting at ERs are in a suicide crisis (ideation specifically)? How many people actually show up voluntarily before an attempt? Does that happen? This is important because a suicidal crisis would be treated differently in the ER if the attempt had already occurred compared to screening for those who might be suicidal. This information would provide a context for the qualitative study.
- Related to the above, how can an ER professional discern in a short time period between physical distress and psychological distress? It seems they would attribute psychological distress to the occurrence of physical distress.
- Was there any mention of referrals to a mental health care specialist outside of the ER? It seems that if there are mental health services available in the community, then help-seekers and support people to the ER would best be served with an adequate mental health assessment followed by referral to an outside source of professional help. However, if there are no resources available outside of the ER, then ER staff need to be trained in dealing with psychological distress.
- It might be helpful to include a summary table of recommendations for improving suicidal distress presentation at the ER and policy changes needed to address the issues.
Maybe there is a quote missing in the title around “Someone Who Can Help”
Reviewer 2 Report
A very actual topic and espcially interesting in these days, the paper is well written and it includes interesting information. There are a few comments to make:
The abstract needs to be improved, especially the background (more focused in the tool than in the topic) and the brief description of the Methods.
There is a global suggestion to write the paper in third-person, not in first person ("We...").
There is no explanations about the aims of the study into the text, we can find information in the abstract and themes in Discussion part. Please, include into the text a specific part about aims of the study to make easier the comprehension.
Methods part can be improved with a brief description about Why this kind of analysis has been chosen and the advantages with other methods. The decission about Qualitative analysis/Grounded Theory is the proper one, it needs to be justified in the text.
I suppose there is one reason to split Methods and to add a part of Analysis. In other Qualitative papers, both are included in same part of the text.
Data and codes are very well done, nice job. Especially the description is very interesting. In fact, it´s much more interesting in the text than in the table.
The part about biases and/or limitations of the study seems to be very important, especially female overrepresentations. Anyway, there are no references about possible mistakes/limitations in data collection and how this limitation can decrease the validity of the paper.
References are very well selected, very actual and connected with the topic
